# The Metabolomics Response of *Solanum melongena* L. Leaves to Various Forms of Pb

**DOI:** 10.3390/nano13222911

**Published:** 2023-11-08

**Authors:** Siyu Zhang, Bing Zhao, Xuejiao Zhang, Fengchang Wu, Qing Zhao

**Affiliations:** 1Key Laboratory of Pollution Ecology and Environmental Engineering, Institute of Applied Ecology, Chinese Academy of Sciences, Shenyang 110016, China; syzhang@iae.ac.cn (S.Z.); zhaobing17@mails.ucas.ac.cn (B.Z.); zhangxuejiao@iae.ac.cn (X.Z.); 2University of Chinese Academy of Sciences, Beijing 100049, China; 3State Key Laboratory of Environmental Criteria and Risk Assessment, Chinese Research Academy of Environmental Sciences, Beijing 100012, China; wufengchang@vip.skleg.cn; 4National-Regional Joint Engineering Research Center for Soil Pollution Control and Remediation in South China, Guangdong Key Laboratory of Integrated Agro-Environmental Pollution Control and Management, Institute of Eco-Environmental and Soil Sciences, Guangdong Academy of Sciences, Guangzhou 510650, China

**Keywords:** metabolite, absorption, metabolic pathway, amino acid, antioxidation

## Abstract

Due to activities like mining and smelting, lead (Pb) enters the atmosphere in various forms in coarse and fine particles. It enters plants mainly through leaves, and goes up the food chain. In this study, PbX_n_ (nano-PbS, mic-PbO and PbCl_2_) was applied to eggplant (*Solanum melongena* L.) leaves, and 379 differential metabolites were identified and analyzed in eggplant leaves using liquid chromatography–mass spectrometry. Multivariate statistical analysis revealed that all three Pb treatments significantly altered the metabolite profile. Compared with nano-PbS, mic-PbO and PbCl_2_ induced more identical metabolite changes. However, the alterations in metabolites related to the TCA cycle and pyrimidine metabolism, such as succinic acid, citric acid and cytidine, were specific to PbCl_2_. The number of differential metabolites induced by mic-PbO and PbCl_2_ was three times that of nano-PbS, even though the amount of nano-PbS absorbed by leaves was ten times that of PbO and seven times that of PbCl_2_. This suggests that the metabolic response of eggplant leaves to Pb is influenced by both concentration and form. This study enhances the current understanding of plants’ metabolic response to Pb, and demonstrates that the metabolomics map provides a more comprehensive view of a plant’s response to specific metals.

## 1. Introduction

In recent years, with extensive attention directed towards atmospheric particulate matter, heavy metals (lead, Pb) present in particles have also gained increased scholarly scrutiny [1,2,3]. Pb is a prevalent heavy metal contaminant, which has garnered attention due to their its non-degradability, bioaccumulation, and high toxicity. Studies have shown that various forms of Pb exist in atmospheric particles. Common Pb forms in inhalable and fine particle aerosols include lead oxide (PbO), chloride (PbCl_2_) and sulfate (PbSO_4_), most of which originate from coal burning [2]. Pb forms in the total suspended particulates of road dust encompass lead sulfide (PbS), PbSO_4_, and PbCl_2_ [1]. These Pb particles can directly endanger human health and life through inhalation [4]. They can also enter plants through leaves and roots in the form of dry and wet deposition. This not only affects the growth and development of plants but also indirectly harms human health by entering the food chain and reaching the human body [5,6,7]. Therefore, it is necessary to study the effects of atmospheric Pb particles on plants.

At present, studies have shown that lead can inhibit the growth of plant roots, cause leaf yellowing and wilting, and dwarf plant growth [8]. By exploring its mechanism through metabolism, it was found that lead ion (Pb^2+^) affected the energy metabolism process of plants [8]. Low-molecular-weight metabolites are the end products of gene expression. Thus, alterations in cell metabolite profiles can be used as an effective strategy for evaluating biological activity [9]. Metabolomics, defined as techniques aiming to offer an unbiased, comprehensive qualitative and quantitative overview of metabolites in living organisms [10], has demonstrated its effectiveness as a tool for comprehending how plants respond to and alleviate various stresses at the molecular level [11,12,13]. Pidatala et al. showed that Pb^2+^ had a significant inducing effect on key metabolic pathways, such as glucose metabolism and amino acid metabolism, in vetiver leaves and roots [14]. Li et al. found that glycolysis, purine, pyrimidine, and phospholipid metabolism were the primary metabolic pathways involved in the response of maize roots to Pb^2+^ stress [15]. These studies provide a basis for understanding the molecular mechanism of terrestrial plants’ response to Pb.

However, these articles have studied the exposure of plant roots to Pb ions, and the metabolic response to Pb after leaf exposure has not been reported. Numerous studies have shown that leaf tissue plays a more significant role in absorbing Pb from atmospheric particulate matter than roots [16,17,18]. Taking radish (*Raphanus sativus*) and parsley (*Petroselinum crispum*) as examples, foliar uptake of Pb was 1.6 times higher than root uptake [19]. Near a secondary lead smelter, foliar uptake of Pb by lettuce and ryegrass was 5~8 times higher than root uptake [20]. Previous studies comparing Fe and Fe_3_O_4_, Cu(OH)_2_ and CuSO_4_, and Ag nanoparticles (NPs) and Ag^+^ have demonstrated that different metal forms have distinct effects on plant metabolism [12,21,22]. Notably, Ag NPs induce disruptions in amino acid metabolism, the TCA cycle, and oxidative stress, which are unique to Ag NPs [21]. However, the study of the effects of different forms of Pb on plant metabolism has not received much attention. Therefore, gaining an in-depth understanding of the metabolic response mechanism of crop leaves to various forms of Pb absorption is of great significance for comprehending Pb exposure and assessing Pb risk.

The purpose of this study was to understand how PbX_n_ (nano-PbS, mic-PbO and PbCl_2_) exposure affects metabolites and metabolic pathways in eggplant leaves, and to identify the metabolic differences and causes of the leaves induced by the three types of Pb. The obtained data not only offer a comprehensive understanding of the mechanism of stress response in eggplant leaves, but will also provide essential baseline knowledge for controlling Pb pollution in agriculture.

## 2. Materials and Methods

### 2.1. Materials

Bulk PbO and PbCl_2_ of analytical reagent grade were obtained from Shanghai Macklin Biochemical Co., Ltd. (Shanghai, China). Lead acetate of analytical reagent grade was obtained from Fu Chen (Tianjin) Chemical Reagent Co., Ltd. (Tianjin, China). Thioacetamide (≥98.0%), cetyltrimethylammonium bromine (99%) and EDTA-2Na (≥99.9%) were sourced from Shanghai Aladdin Biochemical Technology Co., Ltd. (Shanghai, China). Nitric acid (HNO_3_, 69~70%, trace metal < 1 ppb) of guaranteed reagent grade was obtained from ANPEL Laboratory Technologies (Shanghai) Inc. (Shanghai, China). Perchloric acid (HClO_4_, 70~72%) and hydrogen peroxide (H_2_O_2_, ≥30%) of guaranteed reagent grade was sourced from Tianjin Kemiou Chemical Reagent Co., Ltd. (Tianjin, China). Ethanol (99.7%) was acquired from Chengdu Kelong Chemical Co., Ltd. (Chengdu, China). Standard substance citrus (GBW10020) and celery (GBW10048) leaves were purchased from the National Sharing Platform for Reference Materials. Ultrapure water (resistivity ≥ 18.2 MΩ*cm) was prepared using a Milli-Q Direct 8 system (Merck Millipore, Darmstadt, Germany). Eggplant (*Solanum melongena* L.) seeds were purchased from Shenyang Kexing Seeds Co., Ltd. (Shenyang, China).

### 2.2. Preparation of PbX_n_ Stock Solution

Details of the synthesis of nanoscale PbS (nano-PbS) can be found in Appendix A. A 1000 mg Pb/L nano-PbS suspension was prepared and uniformly dispersed via ultrasonic treatment using a water bath ultrasonic generator (KQ-300B, Kunshan, China) at 800 W for 3 h. Bulk PbO was ground into large sheets to facilitate dispersion, and then sonicated at 720 W with an immersing probe in an ultrasonic cell disrupter (1800-99, Biosafer, Shanghai, China) for 3 h to achieve a dispersion of 2000 mg Pb/L and to obtain micron-sized PbO (mic-PbO).

Dissolvable Pb salts (PbCl_2_) were used for comparison with the particles. PbCl_2_ was dissolved in distilled water directly with the assistance of ultrasonication to obtain a solution with 100 mg Pb/L concentration. Stock Pb suspensions/solutions were sealed and stored at 4 °C. Plant experiments should be carried out as soon as possible.

### 2.3. PbX_n_ Characterization

PbX_n_ stock suspensions were sonicated at 800 W for 1 h in a water bath ultrasonicator to achieve better dispersion. Then, the stock suspensions were diluted to an experimental concentration of 10 mg Pb/L. For the determination of particle hydrodynamic size and zeta potential (Nano-ZS, Malvern, UK) in nanopure water, solutions of 10 mg Pb/L nano-PbS and mic-PbO were prepared and subjected to bath sonication (KQ-300B, Kunshan, China) at 50 kHz for 30 min before measurements. The following steps were repeated daily during the 5-day PbX_n_ suspension application stage of the plant experiment. PbX_n_ suspensions were filtered through 0.1 µm mixed cellulose ester membranes to remove particles, and were acidified with 5% HNO_3_ before analysis via inductively coupled plasma–mass spectrometry (ICP-MS, NexION 300X, Perkin Elmer, Waltham, MA, USA) to determine the amounts of ionic Pb. The analytical conditions and quality control of ICP-MS are described in Appendix A. Freeze-dried nano-PbS and mic-PbO samples were imaged using a scanning electron microscope (SEM, Gemini 300, Zeiss, Jena, Germany) to characterize their diameter and surface morphology.

### 2.4. Plant Growth and Experimental Design

Eggplant (*Solanum melongena* L.) seeds were cultivated in plastic pots (diameter: 10 cm; height: 9 cm) filled with nutrient substrates (pH: 6.5~6.8; N, P, K ≥ 12 g/kg; organic matter contents ≥ 40%; Si ≥ 0.3 g/kg). Deionized water was sprayed to moisten the nutrient matrix, a preservative film was attached to maintain the temperature and humidity required for seed germination, and the film was punctured to ensure an adequate oxygen supply. Initially, three seeds were planted in each pot, and the healthiest plant was retained after germination. All plants were placed in growth chambers (day/night period: 16 h/8 h; temperature: 25 °C/22 °C; humidity: 45%/65%), and the photosynthetic luminous power density during the daytime period was 14.5 W/m^2^. Plants were watered with tap water (Pb < 0.1 μg/L) every 2 days to maintain the humidity of the nutrient substrates at 40%. After reaching the 7~8 leaf stage (height: ~25 cm), the plants were foliarly exposed to different forms of Pb, respectively. The experiment was conducted over a 5-day application stage. During the exposure state, 3.4 mL Pb suspensions containing 34 μg Pb were evenly added dropwise to the adaxial surfaces of all leaves of each whole plant using a pipettor each day (Appendix A). Prior to amendment, the Pb suspensions were sonicated for 30 min to achieve a stable dispersion. Over the 5-day application stage, the plants were exposed to a total of 170 μg Pb. We stopped applying and harvesting the plants on the sixth day. Deionized water without any forms of Pb was used as the control in the same manner. Each treatment consisted of 12 replicates.

### 2.5. ICP-MS Analysis for Pb Content in Plants

The roots, stems, and leaves of the five eggplants were separated after harvest. The roots and leaves were immersed in a 20 mM EDTA-2Na solution for 10 min to remove the surface metals. The stems were washed with deionized water three times. Plant tissues were dried to a constant weight at 75 °C in an oven (101-3AB, Taisite, Tianjin, China). After grinding the dried tissues, 0.2 g of the tissues sample was digested using 3 mL HNO_3_-HClO_4_ (4:1, *v*/*v*) solutions and 0.5 mL 30% H_2_O_2_ in a dispelling furnace (DB-3AB, Lichen, Jinjiang, China). Subsequently, the digested sample was diluted with 5% HNO_3_ solution to a final volume of 10 mL before analysis. The concentrations of Pb in various tissue samples were determined using ICP-MS. The recovery was determined to be 94% ± 9% [mean ± standard deviation, *n* = 5].

### 2.6. Liquid Chromatography–Mass Spectrometry (LC-MS)-Based Leaf Metabolomics

At harvest, a portion of fresh tissues were quickly frozen in liquid N_2_ and stored at −80 °C for metabolite analysis by LC-MS [23]. We accurately weighed 200 mg (±1%) of sample in a 2 mL EP tube, added 0.6 mL of 2-chlorophenylalanine solution (4 mg/L) methanol (−20 °C), then vortexed for 30 s. The treated sample was added to 100 mg glass beads, placed into the tissue grinder, and ground for 90 s at 55 Hz. A room-temperature ultrasound was performed for 15 min. The sample was centrifuged at 12,000 rpm for 10 min, and 200 μL of the supernatant was taken and filtered through a 0.22 μm membrane and then added to the detection bottle. Each treatment consisted of six replicates. A quality control (QC) sample was prepared by mixing aliquots of all treatment samples, assuming that the QC sample contained a mean concentration of all components present in the samples under investigation. We used these samples for LC-MS detection. The analytical conditions of LC-MS are described in Appendix A.

### 2.7. Statistical Analysis

The relative abundances of the metabolites were analyzed using a supervised partial least-squares discriminant analysis (PLS-DA) and a multivariate analysis, both conducted using MetaboAnalyst 4.0 (https://www.metaboanalyst.ca/, accessed on 16 September 2022). These analyses aimed to normalize the data via log transformation (normalized by sum), enabling better comparability of individual features [24]. Variable importance in projection (VIP) represents the weighted sum of squares in the PLS-DA analysis, signifying the variable’s significance within the entire model [25]. A variable with a VIP > 1 is considered significant [26]. Additionally, univariate statistical analyses (one-way ANOVA) were performed using online resources (http://www.metaboanalyst.ca/, accessed on 20 June 2023) with a threshold value of 0.05. A biological pathway analysis was conducted based on all identified metabolite data using MetaboAnalyst 4.0 [26]. The threshold value for pathway identification was set at 0.1 [26]. Significant differences in Pb content within plant tissues were assessed using a one-way ANOVA followed by a Duncan test, performed with SPSS 24.0 software (SPSS, Chicago, IL, USA). Differences were deemed statistically significant at *p* < 0.05.

## 3. Results and Discussion

### 3.1. Characterization of PbX_n_

SEM results showed that the average diameter of nano-PbS and mic-PbO is 82 ± 18 and 720 ± 257 nm, respectively (Appendix A). Some 10 mg Pb/L PbX_n_ suspensions were characterized in terms of the average hydrodynamic size, zeta potential (ZP), and dissolution. The average hydrodynamic sizes for nano-PbS and mic-PbO were 316 ± 6 and 1144 ± 70 nm, respectively, with corresponding zeta potentials of −23 ± 1 and 20 ± 1 mV, respectively (Appendix A). The nano-PbS- and mic-PbO-released Pb ions reached equilibrium (4.5 ± 0.1 mg/L for nano-PbS, 4.2 ± 0.2 mg/L for mic-PbO) on the fourth day. The release of Pb ions reached a maximum value of 10.2 ± 0.8 mg/L for PbCl_2_ within one day, and remained largely unchanged over 5 days (Appendix A). Therefore, the two Pb particle suspensions not only contain the original Pb particle type, but also contain 45% lead ions in the nano-PbS suspension, and 42% in the mic-PbO suspension.

### 3.2. Pb Absorption in Eggplant

The absorption and position of Pb in leaves were analyzed via ICP-MS. Exposure to PbX_n_ significantly increased the Pb content of eggplant leaves (*p* < 0.05), with the order nano-PbS > PbCl_2_ > mic-PbO (Figure 1). The Pb content upon exposure to nano-PbS (104.5 ± 4.5 μg), mic-PbO (10.4 ± 1.4 μg), and PbCl_2_ (14.7 ± 0.3 μg) was nearly 235-fold, 23-fold, and 33-fold higher than in the control, respectively. The detected Pb in eggplant leaves may be particles or Pb irons released from particles. Herein, we determined the dissolution dynamics of PbX_n_ in particle aqueous solution for up to 5 days. The dissolution rate of nano-PbS and mic-PbO was 44% and 42%, respectively (Appendix A). The dissolution rate of PbCl_2_ was 97~102%, indicating a complete dissolution. Therefore, most of the Pb in nano-PbS treated leaves may be nano-PbS, while most of the Pb in PbCl_2_ treated leaves may be Pb ions. Additionally, eggplant leaves absorbed more nano-PbS than mic-PbO. This could be attributed to the relatively smaller size of nano-PbS (82 ± 18 nm) compared to mic-PbO (720 ± 257 nm), facilitating its easier transport into guard cells or cuticles and movement between mesophyll cells [27]. Compared with the control group (CK), the content of Pb in the stem was essentially the same as that of CK (0.13 ± 0.03 μg) after nano-PbS (0.14 ± 0.04 μg), mic-PbO (0.11 ± 0.08 μg), and PbCl_2_ (0.14 ± 0.07 μg) treatments (Figure 1). After nano-PbS (0.26 ± 0.06 μg), mic-PbO (0.35 ± 0.13 μg), and PbCl_2_ (0.40 ± 0.06 μg) treatments, there was no significant difference in the Pb content in roots compared to CK (0.25 ± 0.05 μg) (Figure 1). Therefore, the results showed that Pb was not transferred from leaves to stems and roots. It is possible that PbX_n_ in leaves did not reach the vascular bundle and thus could not be transported to the roots [27].

### 3.3. Metabolic Response of Eggplant Leaves to PbX_n_

Metabolite profiles of unexposed eggplant leaves were obtained through LC-MS-based metabolomics. A total of 2484 metabolites were detected, and based on retention indices and mass spectra, 472 metabolites were identified using the Metlin identifier. To compare the metabolite profiles of the three Pb species, an unsupervised multivariate method using a principal component analysis (PCA) was performed on the eggplant leaf datasets. The model was not provided with any prior information regarding the sample identities [28]. The grouping of samples in a PCA scores plot is determined by the similarities in their metabolic profiles.

The PCA scores plot (Figure 2a) shows that the metabolite profiles of unexposed and exposed eggplant leaves were separated along the first principal component (PC1), which explained 15.5% of the total variance. This indicates that the metabolite patterns were different for unexposed and exposed eggplant leaves. All samples were within a 95% confidence interval, and the difference was statistically acceptable without outliers. To visualize general grouping information based on treatment, PLS-DA analysis was performed. The score plot (Figure 2b) demonstrates that all PbX_n_ groups were distinctly separated from the controls, explaining 15.5% of the data variation. These results indicate that PbX_n_ altered the metabolic profile of the eggplant leaves. Additionally, we observe that the metabolic changes are PbX_n_-dependent, with PbCl_2_ inducing the most noticeable metabolic changes, followed by mic-PbO, and then nano-PbS.

The responsive metabolites were subsequently isolated based on VIP score > 1. The Venn diagram of 379 metabolites that resulted in the separation of the control and all PbX_n_ treatment groups was shown in Figure 3. Differential metabolites include sugars and sugar alcohols, amino acids and polyamines, nucleotides, lipids, organic acids, fatty acids, and other metabolites such as phenolic acids and alkaloids. Compared with CK, the number of differential metabolites induced by nano-PbS, mic-PbO and PbCl_2_ was 81, 256 and 267, respectively. Among them, organic acids were the main metabolites, accounting for 20%, 21% and 20%, respectively. Organic acids are strong cation chelators, which play an important role in facilitating mineral element uptake and sequestering or excluding toxic metals [29,30,31,32]. Among 38 identical metabolites induced by nano-PbS, mic-PbO and PbCl_2_, sugars and sugar alcohols accounted for 21% (Appendix A). The results showed that all three lead treatments interfered with the carbon metabolism of eggplant leaves. Among the differential metabolites interfered with by mic-PbO and PbCl_2_ treatments, the same differential metabolites accounted for 44%, while among the differential metabolites interfered with by nano-PbS and mic-PbO, nano-PbS and PbCl_2_ treatments, the same differential metabolites accounted for 20% and 16%, respectively. This indicates that the metabolic response mechanisms of eggplant leaves induced by mic-PbO and PbCl_2_ treatments were more similar.

### 3.4. Metabolic Differences Induced by PbX_n_

#### 3.4.1. Amino Acid Metabolism

Exposure to PbX_n_ in eggplant leaves resulted in significant changes in amino acid metabolism (Figure 4 and Figure 5). Amino acids are important primary metabolites, structural units of proteins and peptides, and precursors for the synthesis of other metabolites. Amino acids have multiple functions in plant growth and other biological processes [33,34]. The relative abundance of some amino acids (glutamic acid, tyrosine, kynurenine, proline, sarcosine, tryptophan, phenylalanine, valine, threonine, glutamine, histidine and γ-aminobutyric acid) was significantly reduced under all three Pb treatments (*p* < 0.05) (Figure 4), indicating that these amino acids were either reduced in synthesis or increased in utilization in response to Pb treatment.

Nano-PbS, mic-PbO and PbCl_2_ induced significant down-regulation of glutamate and tyrosine (*p* < 0.05) (Figure 4). Glutamate is known to be a signaling molecule that plays a role in the antioxidant defenses of plants [35]. The down-regulation of glutamate indicates that the three lead treatments interfered with the signal transduction of leaves. Mic-PbO and PbCl_2_ down-regulated phenylalanine (*p* < 0.05). Phenylalanine and tyrosine, two aromatic amino acids, are central molecules of plant metabolism. They are not only the essential components of protein synthesis, but also important precursors of the shikimate pathway and phenylpropanoid pathway, which produce a large number of secondary metabolites, including phenolic compounds, flavonoids, alkaloids and indoleacetic acid. These secondary metabolites play an crucial role in plant defense and detoxification [36]. The down-regulation of phenylalanine and tyrosine suggests that the shikimic acid and phenylpropionic acid pathways in leaves may have been disturbed. As a product of glutamic acid, the levels of proline in leaves were significant decreased after exposure to nano-PbS and PbCl_2_ (Figure 4). Proline is well known to function as an osmolyte, radical scavenger, macromolecule stabilizer, and metal chelator under heavy metal stress [35,37]. The decrease in proline may be the result of its use as a free radical scavenger or metal-chelating agent. As a branched-chain amino acid, valine is the main constituent of the transmembrane region of membrane proteins [38]. The treatment of mic-PbO and PbCl_2_ significantly down-regulated valine, leading us to speculate that the physiological function of the leaf cell membrane may have been disturbed.

PbCl_2_ treatment also significantly interfered with glutamine, histidine, and γ-aminobutyric acid (GABA), which were not disturbed by nano-PbS and mic-PbO treatments (Figure 4). Glutamine is a nitrogen-rich amino acid involved in nitrate and ammonia assimilation in plants [39,40]. The decrease in glutamine indicates that the ability of plants to obtain nitrogen compounds is reduced. Positively charged histidine was also significantly reduced (*p* < 0.05). Histidine is an essential amino acid for plant cell growth and development, so its reduction reveals the effect of Pb treatment on eggplant growth. In addition, histidine is involved in deamination, suggesting changes in this process as well [34,41]. GABA is a non-protein amino acid that plays an important role in signal transduction, pH regulation, nitrogen storage, plant development and stress defense [42]. The specific down-regulation of GABA by PbCl_2_ indicates that PbCl_2_ induced more severe oxidative stress on the leaves. In general, amino acid levels exhibited a downward trend with all three Pb treatments, indicating that nitrogen metabolism was disrupted.

#### 3.4.2. Fatty Acids Metabolism

Nano-PbS, mic-PbO and PbCl_2_ induced significant down-regulation (*p* < 0.05) of palmitic acid (Figure 5 and Appendix A). Stearic acid was significantly down-regulated by mic-PbO and PbCl_2_, while oleic acid was significantly up-regulated by nano-PbS and significantly down-regulated by mic-PbO (*p* < 0.05) (Appendix A). Bovinic acid and linoleic acid were specifically up-regulated by mic-PbO, linolenic acid was specifically down-regulated, and arachidonic acid was specifically up-regulated by PbCl_2_. Fatty acids are the main component of the plasma membrane. The down-regulation of saturated fatty acids (palmitic acid, stearic acid and caprylic acid) and the up-regulation of unsaturated fatty acids (oleic acid, bovinic acid, linoleic acid and arachidonic acid) caused by PbX_n_ treatment resulted in high levels of unsaturated lipid membrane. The plant plasma membrane may be regarded as the first “living” structure that is the target of metal toxicity [43]. Excessive metal can cause plasma membrane dysfunction, including increased cell leakage [43]. Changes in fatty acid levels may lead to changes in membrane fluidity and permeability, and can be used as a mechanism for stress adaptation. Up-regulation or down-regulation of fatty acids may change plasma membrane composition, thereby changing metal homeostasis [43].

#### 3.4.3. Sugars Metabolism

We found that all three Pb-treatments significantly down-regulated (*p* < 0.05) the levels of cellobiose and raffinose. PbCl_2_ significantly down-regulated xylose (Figure 5 and Appendix A). Since cellobiose, raffinose and xylose are polysaccharides located on the cell wall, the decrease in cellobiose and raffinose indicates that plant cell walls are broken down to produce small sugar molecules, such as osmolytes, to protect cell membranes and plant proteins [44]. D-ribose is the main component of RNA and the raw material for maintaining the energy (ATP and NADH) required for the normal physiological activities of cells. PbCl_2_ induced its significant up-regulation to maintain the energy metabolism of cells. After Pb treatment, the levels of erythritol, ribitol, and arabitol changed (Appendix A). These metabolites are sugar polyols in reduced forms of aldoses and ketoses. Sugar polyols have been reported to act as osmoprotective agents. The accumulation of sugar polyols may help maintain cell hydration levels and cell function [44]. Therefore, changes in the compositions of sugar and sugar polyols may be a protective mechanism for eggplant leaves in response to PbX_n_.

#### 3.4.4. Nucleotide Metabolism

Exposure to PbX_n_ changed nucleotide metabolism in eggplant leaves (Figure 5 and Appendix A). The metabolism of both purine and pyrimidine derivatives was affected. For example, a significant increase in cytidine monophospate (CMP) was accompanied by a decrease in cytosine, indicating that pyrimidine metabolism was disturbed [45,46]. Similarly, guanine was reduced in all three Pb-treatments, indicating that purine metabolism also changed [45,46]. This decrease in nucleobases may be related to the accumulation of nucleotides caused by Pb-treatments. The abundance of other nucleoside, such as guanosine, thymidine and uridine, increased significantly under PbCl_2_ treatment. Compared with nano-PbS and mic-PbO treatments, PbCl_2_ treatment had a more serious effect on nucleic acid metabolism in eggplant leaves.

#### 3.4.5. Organic Acids and Antioxidants

The three Pb-treatments up-regulated the content of several organic acids, such as citric acid, cis aconitic acid and succinic acid, and down-regulated malic acid (Appendix A). These organic acids are key metabolic components in the tricarboxylic acid cycle (TCA cycle) [47]. They are usually related to heavy metal stress and are considered to be reliable indicators of metal accumulation [48]. The accumulation of organic acids can participate in the detoxification process of leaves by chelating metal ions. The up-regulation of TCA cycle intermediates may indicate the activation of TCA pathway. The TCA cycle is the core of the cellular respiration mechanism. Plants with up-regulated respiratory regulation may be to produce the energy required for defense compounds to cope with oxidative stress [49,50]. Another explanation for the up-regulation of citric acid is that plants try to chelate excess metal ions with citric acid [51], which may also be the reason why citric acid is only significantly up-regulated by PbCl_2_ (*p* < 0.05). Nicotinic acid is a key component of the plant pathways involved in redox homeostasis and stress signaling [52]. Nano-PbS, mic-PbO and PbCl_2_ treatments significantly down-regulated the level of nicotinic acid, indicating that Pb induced oxidative stress in eggplant leaves. Ascorbate, salicylic acid and 2-hydroxycinnamic acid, as non-enzymatic antioxidants, were up-regulated by nano-PbS, mic-PbO and PbCl_2_ [53], while 4-hydroxybenzoic acid, 4-hydroxycinnamic acid and shikimic acid were down-regulated (Appendix A). It is speculated that the changes in these secondary metabolites may be related to the non-enzymatic antioxidant defense system.

### 3.5. Biological Pathways Analysis

A more detailed analysis of the relevant pathways and networks altered by nano-PbS, mic-PbO and PbCl_2_ was performed. Pathway enrichment analysis showed that nano-PbS, mic-PbO and PbCl_2_ disturbed 67, 83, and 83 biological pathways in eggplant leaves, of which 9, 7, and 10 pathways were significantly disturbed (*p* < 0.05), respectively (Figure 6 and Appendix A). Among them, galactose metabolism; phenylalanine, tyrosine and tryptophan biosynthesis; and ABC transporters were significant affected by all the Pb treatments, and are related to carbohydrate metabolism, amino acid metabolism and membrane transport, respectively. Two pathways were perturbed by both nano-PbS and mic-PbO: phenylpropanoid biosynthesis and tyrosine metabolism. Phenylalanine metabolism was perturbed by nano-PbS and PbCl_2_. Although all three Pb-treatments perturbed galactose metabolism, nano-PbS treatment only perturbed 11% of metabolites in the pathway, while mic-PbO and PbCl_2_ induced 24% of metabolites, and the metabolites induced by the two treatments were exactly the same. Similarly, although the nano-PbS treatment had the same interference pathways as the other two treatments, the number of metabolites disturbed by nano-PbS was significantly less than that of mic-PbO and PbCl_2_. The results showed that the mic-PbO and PbCl_2_ treatments had a greater impact on related metabolic pathways than the nano-PbS treatment.

Under the stress of nano-PbS, glycine, serine and threonine metabolism, the pentose phosphate pathway and glutathione metabolism were perturbed specifically. Among these pathways, pyruvic acid, betaine, and ascorbate were significantly effected (*p* < 0.05) (Appendix A). Pyruvate is a precursor of branched-chain amino acids, the up-regulation of pyruvate indicates that branched-chain amino acid-related metabolism is activated. Branched-chain amino acids have been shown to be an energy source for oxidative phosphorylation during under adverse conditions in plant [54]. Betaine acts as an osmoprotectant in abiotic stress, maintains plant turgor, stabilizes and protects cell membranes, and also acts as an active oxygen scavenger [55,56]. Ascorbate, as the main metabolite in the glutathione metabolic pathway, acts as a reactive oxygen species scavenger to protect plants from oxidative damage. As an unsaturated fatty acid on the plasma membrane, mic-PbO exposure specifically up-regulated the concentration of linoleic acid, bovinic acid, and 8,10,14-eicosatrienoic acid (Appendix A). This increase in unsaturated membrane lipids plays a essential role in maintaining the necessity of fluidity for membrane function. Aminoadipic acid is an intermediate in the lysine degradation pathway. Lysine degradation in plants is an over-regulated metabolic pathway that effectively converts lysine into glutamate and other metabolites to cope with stress and certain developmental processes [57].

PbCl_2_ specifically perturbed two amino acid metabolic pathways (alanine, aspartate and glutamate metabolism; tryptophan metabolism), two carbohydrate metabolic pathways (butanoate metabolism; TCA cycle), one nucleotide metabolic pathway (pyrimidine metabolism), and a pathway of translation in genetic information processing (aminoacyl-tRNA biosynthesis). PbCl_2_ specifically induced the down-regulation of glutamine, histidine and GABA (Figure 4). Their specific down-regulation indicated that PbCl_2_ had more serious interference on amino acid metabolism in eggplant leaves than nano-PbS and mic-PbO. Glutamine and histidine are also important intermediates in the biosynthesis of amino acid-tRNA. The interference of PbCl_2_ in the translation pathway indicates that the process of tRNA recognizing codons and transporting amino acids is disturbed. Succinic acid and citric acid were specifically up-regulated by PbCl_2_, and fumaric acid was down-regulated. They are important metabolites in butyric acid metabolism and TCA cycling (Appendix A). Cytidine, thymidine, and guanosine were specifically up-regulated by PbCl_2_ (Appendix A). Up-regulation of pyrimidine metabolism may be a strategy for plants to redistribute energy and resources to other stress-related processes [58].

### 3.6. Effects of PbXn Properties on Metabolomics

Our metabolomic analysis showed that PbCl_2_ and mic-PbO induced the amount, degree and metabolic pathway of differential metabolites far more severely than nano-PbS. The differential metabolites induced by mic-PbO and PbCl_2_ were three times that of nano-PbS. However, the absorption of PbCl_2_ and mic-PbO in eggplant leaves was much less than that of nano-PbS, and the absorption of nano-PbS in leaves was ten and seven times that of mic-PbO and PbCl_2_, respectively (Figure 1). Huang et al. studied the antioxidant-related metabolites of cucumber using nano-Cu pesticides, and found that the concentration of antioxidants changed with the dose [59]. Compared with a low concentration (400 mg/L) of lead nitrate, a high concentration (1200 mg/L) of lead nitrate induced more significant up-regulation and down-regulation of metabolites in *Chrysopogon zizanioides* L. Nash [14]. Zhang et al. found that in addition to inducing the same metabolites as Ag^+^, Ag NPs also specifically induced carbazole, raffinose, lactulose, citraconic acid, aspartic acid and other metabolites related to antioxidant defense in cucumber leaves [60]. Liu et al. also found that no specific metabolites in freshwater algae Poterioochromonas malhamensis were only affected by Ag NPs. However, compared with Ag^+^, some metabolites related to amino acid and pyrimidine metabolism were more accumulated or consumed after exposure to Ag NPs [21].

In our results, the differential metabolites induced by mic-PbO and PbCl_2_ were more similar compared to nano-PbS. This may be due to the superposition of ion effects and nano effects. The applied mic-PbO partially entered the leaves in the form of Pb ions (4.2 ± 0.2 mg/L Pb ions), resulting in a 44% overlap in the same metabolites induced by mic-PbO and PbCl_2_. Mic-PbO (1144 ± 70 nm) with a larger particle size can induce the metabolic response of leaves through stomatal entry [61]. This is also the reason that although mic-PbO and PbCl_2_ induced 44% of the same metabolites, they did not induce the same metabolic pathways (except for the metabolic pathway induced by the three kinds of Pb). The uptake of nano-PbS in leaves was ten times that of mic-PbO, but the number of induced metabolites was only one third of that in mic-PbO. Since the concentration of lead ions released from nano-PbS and mic-PbO is similar, we suspect that this may be related to the unique size of nanoparticles. It has been confirmed that metal nanoparticles can penetrate cell membranes [62]. After nano-PbS enters cells, it accumulates in vacuoles to avoid damaging cells due to the cell’s detoxification mechanism [29]. Sulfur ions released by PbS are involved in the detoxification process of cells. Some sulfur-containing substances in plants, such as glutathione, cysteine, and plant-chelating peptides, play an important role in the process of plant resistance to heavy metal stress [63]. It has been documented that under Cu stress, supplying sufficient sulfate or elemental sulfur increases the content of glutathione and phytochelatins in high castors, thereby alleviating stress [64].

## 4. Conclusions

In-depth understanding of the interaction between heavy metal particles and plants is the basis for the prevention and control of heavy metal pollution in agriculture. Three kinds of lead (nano-PbS, mic-PbO and PbCl_2_) were applied to eggplant leaves for 5 days to study their effects on plant metabolomics. The results provide a comprehensive perspective on the molecular changes caused by Pb exposure in leaves. Importantly, although the absorption of nano-PbS by leaves was the largest, the observed metabolic changes were the least, which highlighted the importance of applying heavy metal forms. In eggplant leaves, nano-PbS, mic-PbO, and PbCl_2_ all induced reprogramming of carbon and nitrogen metabolism (including sugars, organic acids, amino acids and nitrogen-containing compounds). PbCl_2_ specifically induced more amino acid metabolic pathways and nucleotide metabolic pathways. This may be due to the high concentrations of Pb ions. Notably, this response might have adverse effects on crop yield and quality during prolonged exposure to Pb-containing particles due to the continuous release of lead ions.

## Figures and Tables

**Figure 1 nanomaterials-13-02911-f001:**
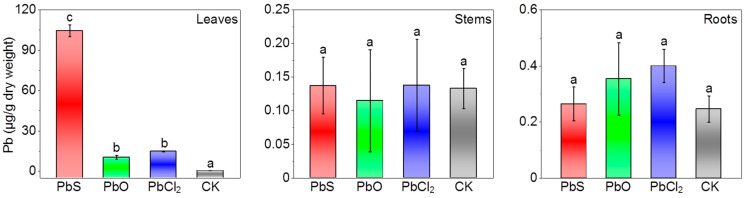
Mass of Pb in the leaves, stems and roots of eggplant plants exposed to PbX_n_ (*M*_Pb_ = 170 μg) for 5 days. The lowercase letters on top of columns indicate significant differences between different Pb species (*p* < 0.05). Data are reported as the mean and standard deviation (*n* = 5). PbS: nano-PbS; PbO: mic-PbO; CK: control.

**Figure 2 nanomaterials-13-02911-f002:**
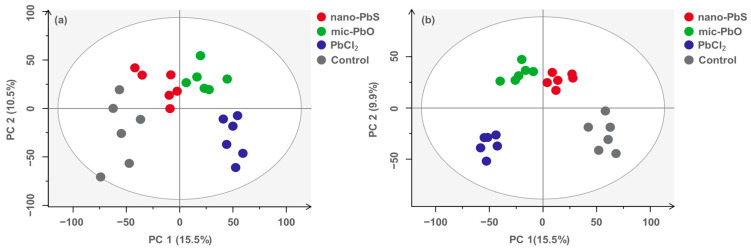
Multivariate analysis of LC-MS metabolite data of eggplant leaves. (**a**) Score plot of the principle component analysis (PCA); (**b**) score plot of the partial least-squares discriminant analysis (PLS-DA). Each treatment had six replicates.

**Figure 3 nanomaterials-13-02911-f003:**
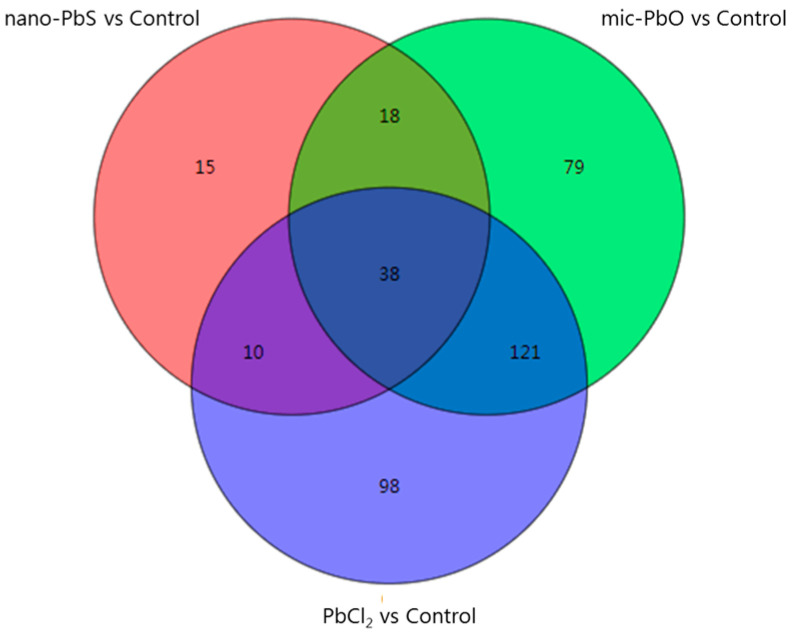
Venn diagrams showing the overlapping and interconnection of metabolites with a VIP score > 1 in eggplant leaves with different PbX_n_ exposures compared with the control. Numbers represent the number of differential metabolites.

**Figure 4 nanomaterials-13-02911-f004:**
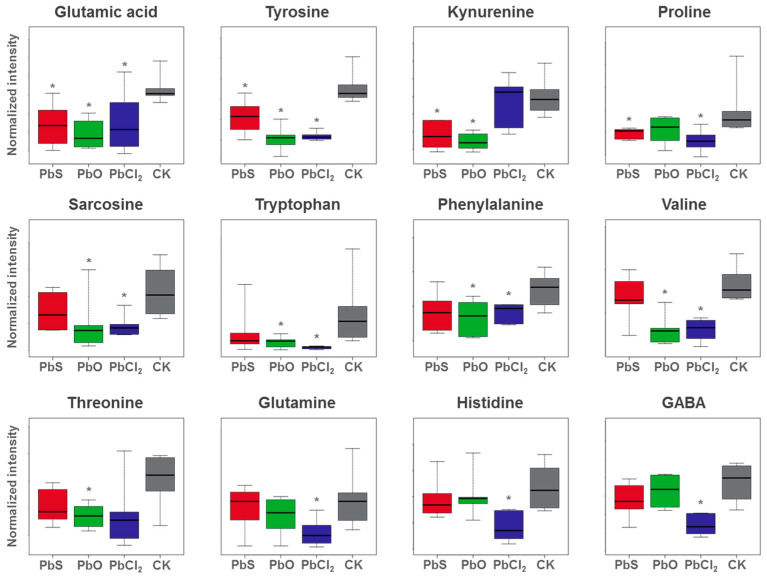
Box plots of the relative intensity of amino acids in eggplant leaves after 5 days of PbX_n_-treatments. PbS: nano-PbS; PbO: mic-PbO; CK: control. * represents a significant difference at the *p* < 0.05 level.

**Figure 5 nanomaterials-13-02911-f005:**
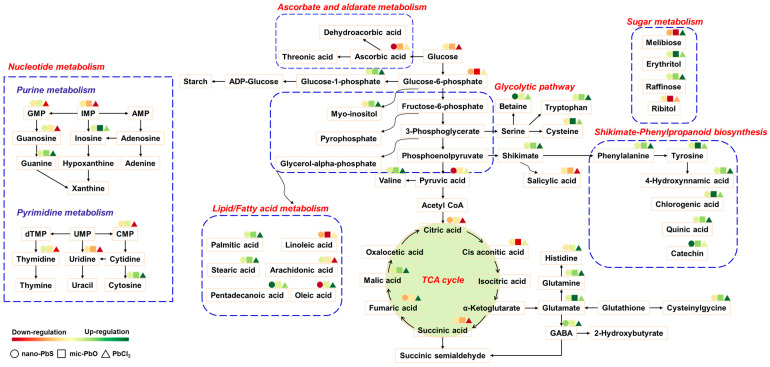
Schematic diagram of the proposed metabolic pathways in eggplant leaves exposed to PbX_n_. Red and green represent the up- and down-regulated metabolites, respectively.

**Figure 6 nanomaterials-13-02911-f006:**
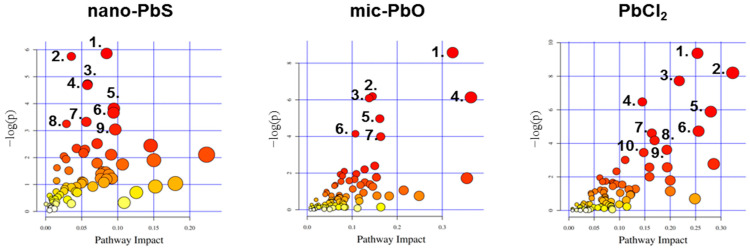
Summary of pathway analysis with MetaboAnalyst 4.0. All identified metabolites were considered in the pathway analysis. The numbers (1–10) represent the metabolic pathways that interferes.

## Data Availability

The data can be shared upon contact with the corresponding author (zhaoqing@iae.ac.cn).

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
