# Peer review of "The Metabolomics Response of Solanum melongena L. Leaves to Various Forms of Pb"

_nanomaterials, 2023, doi:10.3390/nano13222911_

Round 1

Reviewer 1 Report

Studying the influence of heavy metals on plant metabolism is important. This is due to both their influence on the growth and productivity of crops and their subsequent entry into the human body. In the latter case, they can have a toxic effect, which negatively affects health.

Lead is one of the most common representatives of heavy metals that negatively affect the entire body. The authors investigated its effect on eggplant plants. The priority of the work is its use in various forms (Nano-PbS, mic-PbO and PbCl2).

The work was carried out at a good methodological level, time-based research methods were used. A good statistical analysis of the data was carried out.

 All comments are noted in the text of the manuscript.

Recommendations:

The last paragraph of the introduction section (the purpose of the study) needs to be improved. The first two sentences present similar problems.

The method reports analysis of 6-day plants after treatment, whereas the entire text and conclusion refer to 5-day exposure.

Experimental results should be placed in the text of the article, and not just in Supplement.

Conclusions should be consistent with the data obtained, and not exceed them.

Many of them are not confirmed by experimental data.

The stress response to various forms of lead has not been studied. Therefore, this thesis can only be an assumption.

The presentation of the text in English requires correction

Author Response

Responses to Reviewer 1:

Recommendations:

The last paragraph of the introduction section (the purpose of the study) needs to be improved. The first two sentences present similar problems.

Answer: Thank you for the recommendations. We have improved the first two sentences of the last paragraph of the introduction. The revised content is as follows:

“The purpose of this study was to understand how PbXn (nano-PbS, mic-PbO and PbCl2) exposure affects metabolites and metabolic pathways in eggplant leaves, and to identify the metabolic differences and causes of the leaves induced by the three types of Pb.”(Page 2)

The method reports analysis of 6-day plants after treatment, whereas the entire text and conclusion refer to 5-day exposure.

Answer: Thank you for the comment. In 2.4. Plant growth and experimental design, our description of the experimental duration is inaccurate. Lead was applied to the leaves of eggplant for 5 days, and the plants were harvested on the 6th day. The revised content is as follows:

“Stop applying and harvest the plants on the 6th day.” (Page 3)

Experimental results should be placed in the text of the article, and not just in Supplement.

Answer: Thank you for the recommendations. We put Figure S4 in the manuscript Figure 1, as shown below:

 Figure 1. Mass of Pb in the leaves, stems and roots of eggplant plants exposed to PbXn (MPb = 170 μg) for 5 days. The lowercase letters on top of columns indicate significant differences between different Pb species (p < 0.05). Data are reported as the mean and standard deviation (n = 5). PbS: nano-PbS; PbO: mic-PbO; CK: control. (Page 5)

Conclusions should be consistent with the data obtained, and not exceed them. Many of them are not confirmed by experimental data.

Answer: Thank you for the comment. We have modified the conclusions that have not been experimentally confirmed in the manuscript. The specific modifications are as follows :

Modify “This study enhances the current understanding of plants' metabolic response to Pb and demonstrates that the metabolomics map provides a more comprehensive view of plant response to stress factors.” to “This study enhances the current understanding of plants' metabolic response to Pb and demonstrates that the metabolomics map provides a more comprehensive view of plant response to specific metals.” (Page 1)

Modify “... indicating that these amino acids were either reduced in synthesis or increased in utilization in response to Pb treatment induced stress.” to “... indicating that these amino acids were either reduced in synthesis or increased in utilization in response to Pb treatment.” (Page 7)

Modify “The down-regulation of phenylalanine and tyrosine indicates that the shikimic acid and phenylpropionic acid pathways in leaves were disturbed.” to “The down-regulation of phenylalanine and tyrosine suggests that the shikimic acid and phenylpropionic acid pathways in leaves may have been disturbed.” (Page 8)

Modify “It was significantly down-regulated by mic-PbO and PbCl2, indicating that the physiological function of cell membranes may be disturbed.” to “The treatment of mic-PbO and PbCl2 significantly down-regulated valine, leading us to speculate that the physiological function of the leaf cell membrane may be disturbed.”(Page 8)

Modify “The absence of changes in these secondary metabolites indicates that three Pb-treatments trigger non-enzymatic antioxidant defense systems to protect plants from oxidative damage by scavenging reactive oxygen species.” to “It is speculated that the changes in these secondary metabolites may be related to the non-enzymatic antioxidant defense system.” (Page 10)

All comments noted in the text of the manuscript:

Comments in the manuscript text that reiterate the aforementioned recommendations are no longer reiterated.

Text editing required: “Mic-PbO and PbCl2 induced more identical metabolite changes, but the alterations in metabolites related to the TCA cycle and pyrimidine metabolism, such as succinic acid, citric acid and cytidine, were specific to PbCl2.”

Answer: Thank you for the comment. Modify the sentence to “Compared with nano-PbS, mic-PbO and PbCl2 induced more identical metabolite changes. But the alterations in metabolites related to the TCA cycle and pyrimidine metabolism, such as succinic acid, citric acid and cytidine, were specific to PbCl2.” (Page 1)

Text editing required: “... heavy metals (lead) present in particles have also gained increased scholarly scrutiny[1-3]. Lead (Pb) is a prevalent heavy metal contaminant, which ...”

Answer: Thank you for the comment. Modify the sentence to “... heavy metals (lead, Pb) present in particles have also gained increased scholarly scrutiny[1-3]. Pb is a prevalent heavy metal contaminant, which ...” (Page 1)

Text editing required: “At present, studies have shown that lead can inhibit the growth of plant roots, causing yellowing and wilting of leaf surface and dwarfing of plant growth[8].

Answer: Thank you for the comment. Modify the sentence to “At present, studies have shown that lead can inhibit the growth of plant roots, cause leaf yellowing and wilting, and dwarf plant growth[8].” (Page 2)

Repeat with the above text: “By exploring its mechanism through metabolism, it was found that Pb2+ affected the energy metabolism process of plants, thereby inhibiting the growth and development of plants[8].”

Answer: Thank you for the comment. Modify the sentence to “By exploring its mechanism through metabolism, it was found that lead ion (Pb2+) affected the energy metabolism process of plants[8].” (Page 2)

Text editing required: “Thus, alterations in cell metabolite profiles can serve as a potent strategy for evaluating biological activity[9].”

Answer: Thank you for the comment. Modify the sentence to “Thus, alterations in cell metabolite profiles can be used as an effective strategy for evaluating biological activity[9].” (Page 2)

This abbreviation needs to be deciphered: “Ag NPs”

Answer: Thank you for the comment. The “Ag NPs “ was modified to “Ag nanoparticles (NPs)” (Page 2)

“Eggplant (Solanum melongena L.) seeds were cultivated in plastic pots (diameter: 10 cm; height: 9 cm) filled with nutrient substrates (pH: 6.5 ~ 6.8; N, P, K ≥ 12 g/kg; organic matter contents ≥ 40%; Si ≥ 0.3 g/kg).” The seeds are placed in containers and the plants are grown. Correct the text.

Answer: Thank you for the comment. We give a more detailed description of seed germination as follows: “Eggplant (Solanum melongena L.) seeds were cultivated in plastic pots (diameter: 10 cm; height: 9 cm) filled with nutrient substrates (pH: 6.5 ~ 6.8; N, P, K ≥ 12 g/kg; organic matter contents ≥ 40%; Si ≥ 0.3 g/kg). Spraying deionized water to moisten the nutrient matrix, attaching a preservative film to maintain the temperature and humidity required for seed germination, and puncturing the film to ensure an adequate oxygen supply.” (Page 3)

3.1. Characterization of PbXn Nano-PbS and mic-PbO released Pb ions reached equilibrium (4.5 ± 0.1 mg/L for nano-PbS, 4.2 ± 0.2 mg/L for mic-PbO) on the 4rd day. The release of Pb ions reached a maximum value of 10.2 ± 0.8 mg/L for PbCl2 within one day and remained largely unchanged over 5 days (Figure S3).” Differences in the supply of metal present in different forms should be discussed.

Answer: Thank you for the comment. We add a discussion of the differences in the supply of metal present in different forms after this passage. Add the sentence as follows: “Therefore, the two Pb particle suspensions not only contain the original Pb particle type, but also contain 45 % lead ions in the nano-PbS suspension and 42 % in the mic-PbO suspension.” (Page 4)

3.2. Pb absorption in eggplant ... suggesting no translocation of Pb from leves to stems and roots.” Translocation was not studied.

Answer: Thank you for the comment. We have increased the study of transposition, as described below: “Compared with the control group (CK), the content of Pb in the stem was essentially the same as that of CK (0.13 ± 0.03 μg) after nano-PbS (0.14 ± 0.04 μg), mic-PbO (0.11 ± 0.08 μg), and PbCl2 (0.14 ± 0.07 μg) treatments (Figure 1). After nano-PbS (0.26 ± 0.06 μg), mic-PbO (0.35 ± 0.13 μg), and PbCl2 (0.40 ± 0.06 μg) treatments, there was no significant difference in the Pb content in roots compared to CK (0.25 ± 0.05 μg) (Figure 1). Therefore, the results showed that Pb was not transferred from leaves to stems and roots.” (Page 5)

“Figure 3. Box plots of relative intensity of amino acids in eggplant leaves after PbXn-treatments.” How old are the plants?

Answer: Thank you for the comment. Make the following changes to the sentence: “Figure 4. Box plots of relative intensity of amino acids in eggplant leaves after 5 days of PbXn-treatments.”(Page 7)

Reviewer 2 Report

The authors studied PbXn (nano-PbS, mic-PbO and PbCl2) was applied to eggplant (Solanum melongena L.) leaves, and they identified and analyzed 379 differential metabolites in eggplant leaves using liquid chromatography-mass spectrometry. It is quite an interesting manuscript that shows the risks of environmental pollution with heavy metals. The manuscript is well prepared and publishable in its current state.

Author Response

I hope this message finds you well. I am writing on behalf of the authors to express our sincere gratitude for your invaluable contribution as a reviewer for our manuscript titled “Metabolomics response of Solanum melongena L. leaves to var-ious forms of Pb”. Your thorough and thoughtful review significantly enhanced the quality and rigor of our research, and we greatly appreciate the time and effort you dedicated to this process.
